# OpenReview forum: "Shell Games: Control Protocols for Adversarial AI Agents"
_ICLR.cc/2025/Conference — ICLR 2025 Conference Withdrawn Submission_

### Official Review · Reviewer_iXR5 · 2024-10-27

**Soundness:** 3
**Presentation:** 2
**Contribution:** 2
**Rating:** 5
**Confidence:** 2

**Summary:**

The paper explores safety protocols to manage untrusted AI agents completing tasks in a bash shell environment. It introduces the BashBenchHard dataset and develops control protocols that improve both safety and task usefulness by dynamically assessing suspicious commands. The study highlights a "Defer-to-inspiration" protocol, achieving enhanced safety without significant utility loss.

**Strengths:**

1. The introduction of the BashBenchHard dataset provides a realistic, verifiable environment for evaluating AI control in an agentic setting, allowing the blue team to test safety measures against an untrusted AI in more complex, multi-step tasks.

2. The paper develops novel control protocols, including "Defer-to-inspiration," which significantly enhance safety while retaining model usefulness. These protocols offer a balanced approach to managing risk and functionality.


3. The paper includes extensive red-teaming efforts to identify potential attack strategies, improving the robustness of the proposed safety protocols against adversarial actions.

**Weaknesses:**

The protocols assume access to a limited but significant amount of high-quality, trusted human labor for auditing, which may not be feasible in real-world deployments due to high costs and scalability concerns. In control tasks, human auditors are often required to continuously monitor, audit, or verify every interaction or command in real-time, especially in adversarial settings where subtle but dangerous actions can emerge at any step. It seems control tasks are usually complicated and need expertise. It seems hard to implement in real-life scenarios.

**Questions:**

None

---

### Official Review · Reviewer_EL54 · 2024-11-03

**Soundness:** 3
**Presentation:** 3
**Contribution:** 3
**Rating:** 5
**Confidence:** 4

**Summary:**

The paper presents a robust approach for testing and developing control protocols for AI systems that might engage in adversarial behavior. This research emphasizes AI Control as a methodology for creating systems that are secure and resilient against intentional subversion.

**Strengths:**

1. The paper addresses a realistic and pressing problem in AI safety. The setting which considers shell interaction is realistic. Multiple protocols are discussed and evaluated regarding the safety metrics.

**Weaknesses:**

1. The red team’s attack strategies rely heavily on manually designed templates, which may overstate current adversarial capabilities or overlook novel attack vectors that could emerge in unsupervised settings. This dependency might limit the exploration of more sophisticated, emergent adversarial behaviors from advanced AI agents. Is it possible to automatically generate these templates with another agent?

2. The control protocols focus on detecting specific types of suspicious behavior but may struggle to adapt to entirely new forms of attacks. Novel or highly innovative adversarial strategies that do not fit the pre-specified red flags could potentially bypass the current control mechanisms. Are the protocols robust to the dynamic attacks if all details are available to the attacker?

**Questions:**

Please refer to the weaknesses.

---

### Official Review · Reviewer_NVda · 2024-11-03

**Soundness:** 3
**Presentation:** 2
**Contribution:** 1
**Rating:** 3
**Confidence:** 4

**Summary:**

This paper proposes a framework to perform a red-blue team evaluation for certain shell command tasks, using LLMs as the teams and possible monitors, and possibly a small set of human auditors. There is a trusted of low ability and untrusted worker of higher ability (workers are LLMs). The red team aims to run attacks via untrusted worker and the blue team aims to stop any such attack while also extracting useful work from untrusted worker.

**Strengths:**

This is a benchmarks paper and I believe the authors have chosen a primary area that is not suited for this paper. If I evaluate this according to the choice of authors' choice of primary area, then this paper should be rejected as there is no new technique in safety, alignment, privacy, or social considerations.

1) The paper releases a dataset that can be useful.
2) This can be a valuable basis for further engineering.

**Weaknesses:**

1) The paper lacks any principled exploration of optimal strategies - the strategies devised are created based on human intuition, there is no model of the overall interactions and no methodology.
2) The writing in many places is confusing. I am stating one example from the abstract, similar examples appear in many places: "The blue team has access to a weaker trusted model and a limited amount of high-quality, trusted labor to attempt to safely extract work from a stronger untrusted model" - what does it mean to extract work?
3) This works seems an application using LLMs, that too for a very specific task. I do not get what is the research part in this paper?

I am not sure if this paper fits in a top AI research conference.

**Questions:**

None

---

### Official Review · Reviewer_rBEy · 2024-11-04

**Soundness:** 3
**Presentation:** 4
**Contribution:** 3
**Rating:** 5
**Confidence:** 3

**Summary:**

This paper extends prior work on AI Control in an agentic setting where a(n untrusted) model operates within a bash shell to solve tasks designed as part of a benchmark. The evaluation set up consists of a red team (the untrusted model) and blue team (which is the trusted monitoring/control system here), and the paper seeks to evaluate optimal defense protocols based on red team attack strategies in this setting.

**Strengths:**

- The threat model is quite novel and realistic compared to previous work in AI Control
- The experiments are quite large scale in terms of number of tasks on the benchmark
- The paper presents two improved control protocols, trusted inspiration and defer to resampling
- The paper is on an important, underexplored topic in AI safety and can guide future work on monitoring systems and control evaluations

**Weaknesses:**

- Overall, the setting does not appear agentic in a sense. Despite the paper being built around agents acting as bash assistants, it is essentially focused on model outputs directly. The blue team does not have any context over the effects of actions on the bash shell. Looking at only the outputs from the untrusted model appears fundamentally limited, especially in this setting.
- The paper appears limited in novelty, especially in terms of the control protocols it presents compared to [1]. There are two new control protocols presented, trusted inspiration (where the trusted model generates an initial guide for the untrusted model) and defer to resampling, where the monitor asks the untrusted model to generate multiple samples for an answer and can choose/reject accordingly. However, these control protocols don’t appear particularly novel. Other protocols to try out could be: (1) suspicion accumulation over steps and dynamic thresholding based on prior outputs or (2) get the untrusted model to generate an explanation/CoT steps/verifiable steps for its outputs which then gets evaluated or (3) some debate strategy between the monitor and untrusted model.
- The paper states assumptions around schelling points, collusive capabilities, encoded reasoning, etc. that do not reference works such as [2][3][4][5][6]. It would be important to compare the paper's assumptions to experimental results from these works or run initial experiments validating these assumptions.
- No adversarial training strategies tried out or developed for either the untrusted model or monitor. This appears to limit the scope of the paper, since real world applications will rely on trained monitors.

[1] AI Control: Improving Safety Despite Intentional Subversion

[2] Preventing Language Models From Hiding Their Reasoning

[3] https://github.com/openai/evals/blob/main/evals/elsuite/schelling_point/README.md

[4] Secret Collusion among Generative AI Agents

[5] https://github.com/openai/evals/blob/main/evals/elsuite/steganography/readme.md

[6] Generative Text Steganography with Large Language Model

**Questions:**

- Is there a reason that some intuitive control protocols (mentioned in weaknesses) are not tried out in the paper?

- Does there need to be a restriction on the blue team observing only outputs from the untrusted model and not their execution on the shell? Does this limit the setting from being 'agentic'

- Why not get the untrusted model to generate an explanation and CoT steps and get some consistency score or train a small trusted verifier on these explanations?

- Are there any scenarios where the resampling process could be more of a threat than an improvement to safety?

- What happens in settings where the trusted model is completely incapable at monitoring a task/judging it's safety? There could very clearly be constraints on how small of a model can act as a trusted model or scenarios where step jumps are seen in terms of capability increases between two models (the blue and red team here).

Overall, it would be great to see these questions answered and have the weaknesses addressed. If I have misinterpreted or skipped anything, I would be quite happy to raise my score.

---

### Note · Authors · 2024-11-27

**Comment:**

We would like to formally withdraw our paper submission from ICLR 2025.

Since the initial submission, we have made substantial updates and revisions to the work that significantly change its scope and findings. Given these major changes, we believe it would be most appropriate to withdraw the current version rather than proceed with the review process.

We sincerely appreciate the time and effort of everyone involved in the review process, and took the reviews into consideration for our updated paper. We hope to submit the updated version of this work to a future venue.

Thank you for your understanding.

**Withdrawal Confirmation:**

I have read and agree with the venue's withdrawal policy on behalf of myself and my co-authors.